# Dynamics of the Actin Cytoskeleton at Adhesion Complexes

**DOI:** 10.3390/biology11010052

**Published:** 2021-12-30

**Authors:** Nicholas M. Cronin, Kris A. DeMali

**Affiliations:** Department of Biochemistry, Roy J. and Lucille A. Carver College of Medicine, University of Iowa, Iowa City, IA 52242, USA

**Keywords:** actin cytoskeleton, mechanotransduction, force, cadherin

## Abstract

**Simple Summary:**

All cells in the human body experience force. To withstand these forces, cells rearrange and reinforce their cytoskeletons. In this review, we describe the structure of the actin cytoskeleton and its anchorage at points where cells adhere to the extracellular matrix and one another. We examine the current understanding for how the actin cytoskeleton is remodeled by force. This examination reveals that the response of cells to force is complex and highly coordinated.

**Abstract:**

The shape of cells is altered to allow cells to adapt to their changing environments, including responding to internally generated and externally applied force. Force is sensed by cell surface adhesion proteins that are enriched in sites where cells bind to the extracellular matrix (focal adhesions) and neighboring cells (cell–cell or adherens junctions). Receptors at these adhesion sites stimulate intracellular signal transduction cascades that culminate in dramatic changes in the actin cytoskeleton. New actin filaments form, and/or new and existing filaments can be cleaved, branched, or bundled. Here, we discuss the actin cytoskeleton and its functions. We will examine the current understanding for how the actin cytoskeleton is tethered to adhesion sites. Finally, we will highlight recent studies describing how the actin cytoskeleton at these adhesion sites is remodeled in response to force.

## 1. Introduction to the Actin Cytoskeleton

Actin is a monomeric, globular protein (G-actin), and its function is directly coupled to its ability to polymerize into filaments (F-actin) [1,2]. Actin filaments can further assemble into higher order structures, such as bundles and networks (reviewed in Skau and Waterman) [3]. Rapid assembly, disassembly, and reorganization of these structures is critical for cell shape, motility, and force generation to be quickly modulated [4,5,6]. Actin rearrangements are facilitated by molecules, known as actin-binding proteins [5,7,8], which bind and modulate actin dynamics by directly increasing or decreasing the polymerization rate, sequestering G-actin, bundling F-actin, or anchoring F-actin to the plasma membrane.

G-actin can polymerize into filamentous structures known as F-actin via a complex, multi-step process (Figure 1A) [1,9,10]. Polymerization is initiated by the assembly of actin monomers into trimers or tetramers [11]. These small actin oligomers bind actin nucleation factors that accelerate the rate of actin polymerization to facilitate growth of filaments. The formin family of proteins accelerate the formation of linear actin filaments (Figure 1D) [8,12]. In contrast, the Arp2/3 complex accelerates actin polymerization of branched actin filaments by binding to the sides of existing filaments (Figure 1E) [13]. Polymerization is further affected by capping protein, which binds to the barbed ends of actin filaments and blocks the addition of actin monomers (Figure 1F). Actin filament stability can be decreased by actin severing proteins, such as, cofilin (Figure 1C).

Addition of new monomers to growing actin filaments can occur at both ends. The on-rate for the plus end of the filament is approximately ten times faster than the minus end [2]. These differences contribute to the highly dynamic phenomenon known as treadmilling (Figure 1B) [1]. As the filament grows, the ATP associated with the actin monomers is hydrolyzed to ADP. ATP hydrolysis does not affect the polymerization rate, rather it defines its mechanical characteristics of the filament—ATP-bound filaments are rigid, whereas ADP-bound filaments are more flexible [14].

The polymerized F-actin can form higher order structures such as bundles and branched actin networks that provide physical stability to cell shape and modulate biological functions. Bundles of actin are produced through extensive F-actin crosslinking and can be parallel or anti-parallel. Proteins that facilitate bundling include vinculin, α-actinin, fascin, myosin, formin, or fimbrin (Figure 1G) [4]. Parallel, bundled actin filaments are prominent in cultured cells and are anchored to integrins, which mediate the adhesion of cells to the extracellular matrix. They are also prevalent in epithelial and endothelial cells in regions where these cells contact their neighbors (Figure 2A). These bundles form a cortical actin belt that is physically tethered to the cadherin adhesion receptors (Figure 2A) [15]. Anti-parallel bundles, on the other hand, are frequently located in protrusions in migrating cells in regions where finger-like protrusions known as filopodia, form (Figure 2B). In contrast, branched actin filaments are found in regions where cells extend their membranes in flat sheets known as lamellipodia to facilitate cell migration and the formation of nascent cell–cell junctions [7,16]. These networks are formed predominately via activation of the Arp2/3 complex, which binds to the sides of existing actin filaments and nucleates the formation of new actin structures (Figure 2B) [13,17]. Branched actin structures are highly dynamic. Fluorescence recovery after photobleaching (FRAP) studies demonstrated branched networks at the leading edge of cells had recovery rates as fast as 1–2 s^−1^. While bundles generally have a larger half-life of 10–15 s^−1^, their rates of recovery can be modified depending on the availability of certain actin-binding proteins [18]. One well understood actin-binding protein with known roles in modulating actin polymerization is cofilin, which is known to sever actin filaments, thereby creating more positive ends on filament fragments [19].

The actin cytoskeleton is bound to myosin proteins and the movement of these myosin proteins along the actin filaments produces contractility. The non-muscle myosin II family members are critical in the development of force-generation/transmission at adherens junctions [20,21] and focal adhesions [22,23,24]. Of the three non-muscle myosin family members, only non-muscle myosin IIA and IIB have both been reported to localize to adherens junctions [25,26] and focal adhesions [27,28,29]. Recent work demonstrated non-muscle myosin IIA and IIB have unique functions in cell–cell and cell–matrix adhesions as discussed later in this review [30,31]. Thus, the actin cytoskeleton is a complex network of actin molecules and actin-binding proteins, which play key roles in allowing cells to respond to external cues.

In this review, we explore how force affects the actin cytoskeleton. We begin considering how the actin cytoskeleton is anchored at points of cell–matrix and cell–cell adhesion. Next, we will consider how force stimulates actin-binding proteins to reorganization of the actin cytoskeleton. Lastly, we will consider recent data demonstrating how force can directly reshape actin filament formation and higher adhesive structures.

## 2. Force Generation 

### 2.1. Actin Polymerization and Force Generation 

Membrane protrusions, such as lamellipodia and filopodia, are regions of active actin polymerization, and actin polymerization has been proposed to generate intercellular force (Figure 2B). There are two established models that can explain how actin polymerization can generate force: (1) the Brownian ratchet [32] and (2) elastic ratchet [33]. The Brownian ratchet model proposes that the thermal undulation of the plasma membrane allows for the addition of a single monomer. Anchored filaments then allow for the monomer addition to move the plasma membrane forward. The elastic ratchet model suggests that the filament itself is a flexible and not rigid structure, and therefore its own thermal movement allows for a gap to form near the plasma membrane allowing for the addition of a monomer. Some work has been undertaken that demonstrates that the elastic ratchet model explains the force generation needed to propel *Listeria*; however, the elastic model shows less accuracy when explaining lamellipodia formation in migratory cells [33,34].

The amount of force generated by actin polymerization is dependent upon the type of actin structure assembled. For example, the force generated from dense, branched actin structures in lamellipodia has been measured using cantilever deflection techniques to be approximately in the 1–2 nN/cm^2^ range [34]. In contrast, bundled actin networks in filipodia generate force is approximately 3–5 pN. Interestingly, in vitro experiments measured bundled actin force generation to be approximately 0.8 pN, which was surprising given that this is the amount of force reported to be generated by a single actin filament [35]. These discrepancies between in vitro measurements and in vivo measurements have been postulated to be reconciled by the observation that non-muscle myosin and actin-binding proteins facilitate force generation within filipodia in vivo. Indeed, when non-muscle myosin is inhibited, the force generated by filipodia is reduced to the same values obtained in the in vitro studies [36].

The amount of force generated by actin polymerization is also dependent upon the type of protrusion formed. Much of the understanding for how actin polymerization generates force is obtained from studies of adherent cultured cells plated on stiff plastic surfaces. In 3D cultures, actin polymerization drives lamellipodial protrusion to facilitate migration and invasion and actomyosin contractility generates hydrostatic pressure and flow of cortical actin network to form membrane blebs that promote forward movement (reviewed in [37,38]). The forces propelling the cell forward are several orders of magnitude lower than during traditional two-dimensional migration on plastic substrates [39] Therefore, there is clearly more work that is required to mechanistically understand the determinants for how force generation modulates actin polymerization.

### 2.2. Non-Muscle Myosin Force Generation

Increasing actin polymerization is not the only mechanism for increasing intercellular force. Non-muscle myosin II motor proteins generate the bulk of mechanical force within cells. Non-muscle myosin II is a motor protein that can bind to F-actin and produce contraction by pulling actin filaments in opposite directions [40]. This contractility is needed to maintain tension on the cortical actin belt, a dense cluster of F-actin that spans the apical region of epithelial and endothelial cells, which binds to proteins at adherens junctions [41] and to bundled actin filaments in focal adhesions. Given this critical role, non-muscle myosin II is required for many physiological events, such as migration, endocytosis, cell division, and tissue formation [22,42], and its loss leads to inflammation, cardiovascular, and digestive diseases [43].

There are several isoforms of non-muscle myosin II, designated as IIA, IIB, and IIC. Non-muscle myosin IIA is the main force generating myosin isoform and is necessary for generating tension in perijunctional actin bundles at nascent adherens junctions [30]. In contrast, non-muscle myosin IIB is found within the junctional branched network and required for proper formation of nascent adherens junction formation [30]. Similarly, in focal adhesions, non-muscle myosin IIA is at the leading edge of cells and contributes to focal adhesion stability and durotaxis, whereas non-muscle myosin IIB is found at the rear and produces tension on stress fibers generating force to dissociate focal adhesions [42]. Thus, non-muscle myosin isoforms play unique and critical roles in maintaining the intercellular tension necessary for cell–cell and cell–matrix adhesions.

## 3. Focal Adhesions

Sites where cells adhere to components of the extracellular matrix are known as focal adhesions and are comprise of numerous structural and signaling proteins. In focal adhesions, heterodimeric cell surface receptors known as integrins bind to extracellular matrix components such as collagen, fibronectin, and vitronectin. Distinct extracellular matrix components bind to unique integrin heterodimers present [44,45]. There are least 24 unique heterodimeric combinations made up of 18 different α- and eight β-subunits. Teasing apart discrete signaling pathways from integrins has been a challenge because of both the sheer number of combinations of heterodimers and the redundancy in signaling pathways.

Focal adhesions arise from a complex orchestration of integrin engagement, intracellular signaling, and directed actomyosin contractility [46]. An important determinant for focal adhesion growth or maturation is the tensile force generated by integrin engagement. The extracellular domain of integrins transitions between relaxed and tensioned states in response to myosin II-generated cytoskeletal force [47]. The energy required for the conformational changes may be provided by forces generated by the actin filaments inside the cell to which the integrins are bound through adaptor proteins [48]. Indeed, talin and other adaptor proteins, such as kindlin, filamin, and tensin bind directly to the C-terminus of integrins and to F-actin, serving as a bridge between adhesion receptor and the actin cytoskeleton [49]. Talin is one of the most well understood mechanosensitive proteins at focal adhesions. When talin is subjected to tensile force, it unfolds near its rod domain, thereby exposing cryptic vinculin binding sites (Figure 3A,B) [50]. Vinculin recruitment to focal adhesions under increased tension allows for dispersion of force across the vinculin–actin contacts, thereby stabilizing the focal adhesion [51].

Focal adhesion development is facilitated by the recruitment of signaling proteins, such as Src, FAK, and PAK, that initiate downstream signaling events. Focal adhesion kinase (FAK) is one of the first auxiliary proteins recruited to focal adhesions [52] (Figure 3B). FAK supports focal adhesion growth by activating several signaling pathways that converge on increased myosin activity and actin nucleating factors [53]. Recently, it has been proposed that FAK might be activated independently of biochemical signaling from other proteins at focal adhesions [54]. This work proposed that tensile force between the FERM domain and the kinase domain of FAK could release autoinhibition and allow for kinase activity. Such an observation might suggest that the temporal recruitment of FAK to focal adhesions is initiated by the onset of the mechanical load and not by biochemical signaling [54].

## 4. Cell–Cell Junctions

The actin cytoskeleton is also anchored to other cell surface adhesion receptors. For cells that form monolayers, such as epithelial and endothelial cells, there are two major points of adhesion: focal adhesions, which we have already discussed, and sites where cells bind to neighboring cells, known as cell–cell contacts or adherens junctions. There are several components of cell–cell junctions that are classified by their spatial distribution and function. The tight junctions are the most apical and regulate paracellular transport of ions and molecules. Transmembrane proteins, such as occludin, claudins, and junctional adhesion molecules, make up these sites of cell–cell contact [55], and are linked to the actin cytoskeleton through adaptor proteins, such as zonula occludens (ZO) 1–3 and cingulin [56]. The C-terminal tail of ZO proteins bind F-actin, thereby forming a core complex for actomyosin contractility and dynamic rearrangement F-actin at tight junctions [57]. Like ZO proteins, cingulin can also facilitate transmembrane protein interaction and F-actin binding [58].

Immediately beneath the tight junctions, yet still apically located in the cell, are the adherens junctions. The adherens junctions of epithelia are highly enriched in epithelial (E)-cadherin [59]. The extracellular domain of E-cadherin forms homophilic interactions with E-cadherin proteins on the same cell and with E-cadherin proteins on adjacent cells—these interactions are increased upon exposure to force. On the inside of the cell, E-cadherin recruits p120, β-catenin, α-catenin, and vinculin. These adaptor proteins play a role in dissipating the force on the actin cytoskeleton. α-Catenin is the most understood of these proteins—it is recruited to the E-cadherin cytoplasmic tail indirectly via β-catenin; it is mechanically active and undergoes conformational changes that expose binding sites for vinculin [20,60]. Vinculin, also an actin-binding protein, further stabilizes the interaction with E-cadherin and the actin cytoskeleton [61]. Together, these events support the growth of the cadherin adhesion complex and are followed by increased density of F-actin [62]. The increase in F-actin at adherens junction under force is thought to increase the ability for cells to withstand externally applied forces (Figure 4). F-actin bundling and rearrangement at adherens junctions under force is not fully understood, but non-muscle myosin II contractility is required [63]. Myosin contractility, as previously discussed, increases the stability of adhesion complexes by facilitating clustering and signaling for further recruitment of actin-binding and polymerizing proteins. Additionally, at adherens junctions, myosin maintains constant tension on the cortical actin belt allowing for rapid force transmission between adjacent cells.

## 5. Mechanotransduction at Adhesions

Mechanotransduction is highly dependent upon force-stimulated conformational changes in proteins [64] to convert mechanical forces into biochemical signals. The transfer of information is initiated by integrins (in focal adhesions) and cadherins (in adherens junctions). Upon sensing force, the cytoplasmic domains of the alpha and beta-integrins dissociate allowing proteins, such as talin, to bind. Talin is a critical intermediate between integrins and the actin cytoskeleton. Increased tensile force on integrins, either externally or internally derived, leads to conformational unfolding of talin, revealing cryptic binding sites within its rod domain [49,65]. These exposed sites allow for proteins, such as vinculin, to bind (Figure 3B). Furthermore, vinculin can associate with new actin filaments furthering the association of the adhesion complex with the actin cytoskeleton (Figure 3B). A similar process is operational at adherens junctions where force on E-cadherin causes stretch-induced conformational change in α-catenin, revealing cryptic binding sites and allowing vinculin to bind. Interestingly, vinculin is differentially recruited to either site via unique phosphorylation sites [61]. Like α-catenin, vinculin undergoes conformational changes in response to force [66].

In addition to undergoing conformational changes and recruiting force-sensitive proteins, both cadherins and integrins stimulate downstream signal transduction cascades in response to force. The most well studied signaling event downstream of both integrins and E-cadherin is activation of the Rho family of GTPase proteins. The Rho GTPases are a family of 20 signaling proteins that are activated when bound to GTP and inactivated when bound to GDP. Some of the prominent members of this family are RhoA, Rac1, and Cdc42. The activation state of Rho GTPases is regulated through guanine nucleotide exchange factors, which activate these proteins by catalyzing the exchange of GDP for GTP, and GTPase-activating proteins, which inhibit the Rho proteins by stimulating their intrinsic GTPase activity [67].

## 6. How Force Influences Actin Cytoskeletal Dynamics

External force and the ensuing conformational changes in proteins and activation of the Rho family of proteins increases the rigidity of cells [68]. This increase in rigidity is the result of dramatic elevations in both focal adhesion and adherens junction size as well as robust enrichments of the actin cytoskeleton—two responses that are critical for withstanding force and preserving cell integrity.

The mechanisms underlying reinforcement of adhesions and the actin cytoskeleton are emerging. Both integrin- and cadherin-containing adhesions increase in number and size in cells experiencing force [69]. Indeed, E-cadherin is concentrated in regions of cell–cell contacts with the highest tension [69], and force stimulates the growth of cell–cell adhesions [70]. Force also increases adhesion strength [21,71]. The increased robustness of the cadherin adhesion complex requires vinculin, α-catenin, and other linkages to the actin cytoskeleton [71,72]. Together these increases boost cell stiffness [73]. Similarly, integrin-containing adhesions cluster, strengthen, increase their attachment to the underlying actin cytoskeleton in response to force [74,75,76], and grow in the direction of the applied force [77]. Additionally, tyrosine phosphorylation of several focal adhesion components is increased by tension [78]. Thus, adhesions increase in size and strength to withstand force.

Like cellular adhesion, actin enrichment at adhesions sites is increased in response to force [74]. How this actin enrichment arises remains under investigation. Using techniques that allow for the simultaneous measurements of the amount of polymerized actin and the rate of F-actin turnover to be examined, Osborn et al. observed the earliest response of endothelial cells to shear stress is a depolymerization of actin, which is rapidly followed by a robust actin polymerization phase [79]. Consistent with a reorganization of actin in response to force, a recent study by Pandit et al. demonstrated that shear strain on branched actin induces daughter filament debranching [80]. Here, the authors used a microfluidics device to apply pN levels of shear strain to Arp2/3 branched actin and discovered that daughter filament dissociation was substantially increased when compared to the controls. Other work indicates that force affects actin by changing its turnover. A study using isolated actin filaments revealed that F-actin filaments under tensile loads of 20 pN had twofold increased lifetimes when compared to those without tensile force [79]. Interestingly, preconditioned loading of actin filaments with 15–27 pN and then application of tension increased lifetimes by more than 30-fold [81]. Thus, both the amount and lifetime of actin are affected by force. More work is needed to determine which phenomena are functional in vivo.

## 7. Conclusions and Perspectives

While stably anchored at adhesion sites and appearing to be a static structure, the actin cytoskeleton is in fact a very dynamic structure that undergoes robust changes in response to internal and external force. These changes are critical for allowing cells to adapt to their changing environments. Interestingly, emerging evidence indicates that changes in the actin cytoskeleton come at an energetic cost to the cell and metabolism is altered to provide for the energy expenditures required to maintain the cytoskeleton [61,82]. Much more work is needed to better understand the relationship between actin cytoskeletal dynamics and cell metabolism. An examination of whether different metabolic pathways support different actin architectures would provide key insights into diseases that are associated with defective metabolism and cytoskeletal defects. Additionally, it is becoming increasingly important to understand how the actin cytoskeleton responds to different amplitudes of force. Investigations of the physiological response of cells under force have noted that the duration and magnitude of force are key aspects to how cells respond. Recent studies are beginning to provide insight into how low and high amplitudes of force affect actin cytoskeletal dynamics [76,77]. This work and others are beginning to conceptually advance our understanding for how actin assemblies form and are modified in different environments.

## Figures and Tables

**Figure 1 biology-11-00052-f001:**
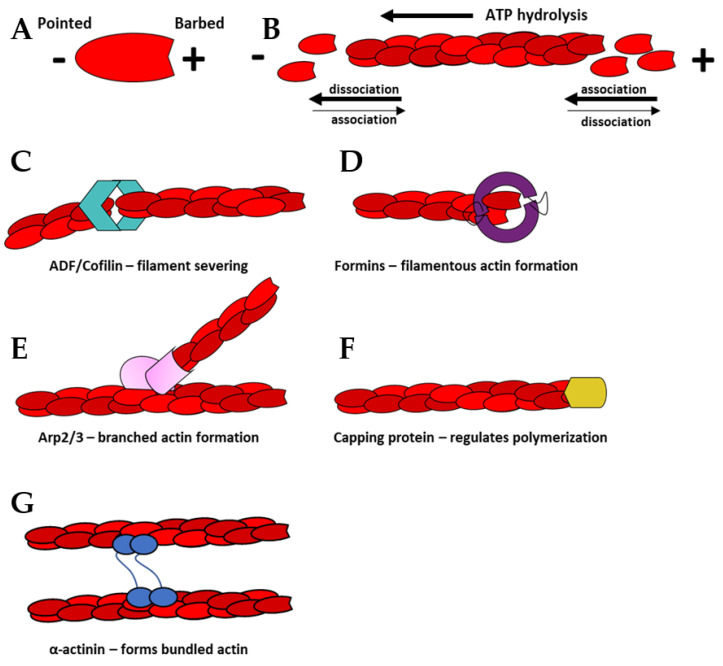
Illustration of actin polymerization and common actin binding or nucleation factors: (**A**) Actin monomers have a barbed (+) and pointed (-) end. ATP binds and magnesium ions bind to the barbed end. Filaments lengthen by binding barbed end to pointed end. (**B**) The monomer association rate is greater at the barbed end of the growing actin filament than the pointed end. This differences in association and dissociation rates at either end of the filament is what leads actin treadmilling. (**C**) ADF/Cofilin, (aqua), severs actin filaments preferentially towards the pointed end. Cofilin therefore, significantly decreases the average lifetime of filaments. (**D**) Formins (purple), increase the rate of polymerization by directly associating with either actin seeds or established filaments and monomeric actin. (**E**) The seven-subunit nucleating complex Arp2/3 (pink), associates with established filaments and allows for the formation of branched networks. (**F**) Capping protein (CP) (yellow) regulates filament lifetimes by preventing filament growth at the barbed end. (**G**) Bundling proteins, such as α-actinin (blue), link parallel actin filaments together.

**Figure 2 biology-11-00052-f002:**
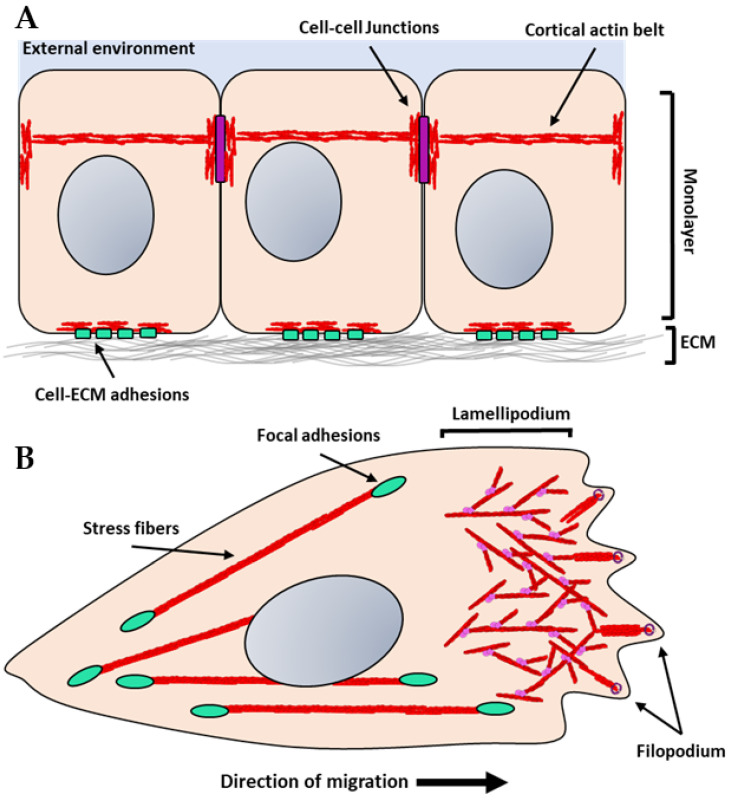
Illustration of actin architectures in monolayers and migratory cells: (**A**) Epithelial or endothelial cells form monolayers that are connected through cell–cell adhesion complexes (purple rectangles). These adhesions are bound to the actin cytoskeleton and form a cortical actin belt. The cortical actin belt links adjacent cells together providing stability and enables mechanical stimuli to be transmitted quickly throughout adjacent cells. These cells also engage with the extracellular matrix (ECM) at the basal interface through adhesion receptors such as integrins (light green). Integrins cluster forming stable focal adhesions, which bind to the actin cytoskeleton through adaptor proteins. (**B**) Migratory cells form extensive branched actin networks towards the leading edge of the cells known as lamellipodia. Small protrusions, known as filipodia, from the leading edge are filled with dense bundled actin. These structures generate protrusive force through both actin polymerization and non-muscle myosin II-mediated contraction. Formins are concentrated in the filopodia (purple), whereas the Arp2/3 complex (pink) is found in lamellipodia. Stress fibers are stable actin bundles that are stabilized by crosslinking proteins such as α-actinin and are generated at focal adhesion sites.

**Figure 3 biology-11-00052-f003:**
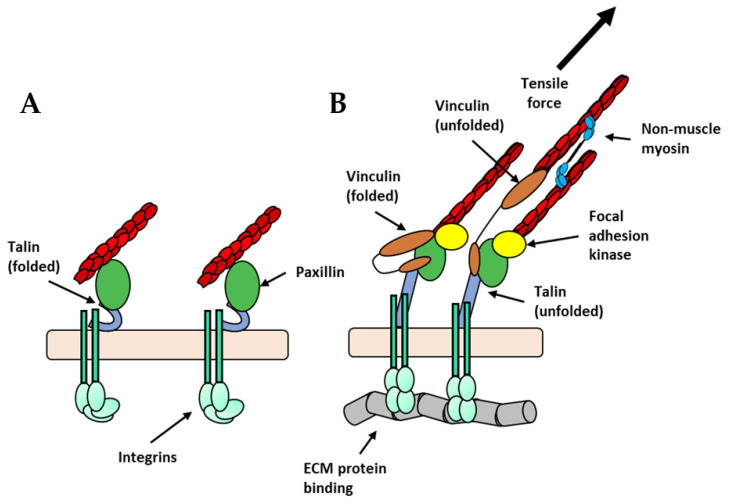
Hypothetical illustration of talin unfolding and growth of focal adhesions: (**A**) In the absence of ECM binding, integrins are in a bent, inactive state (light green), talin is in its folded conformation (grey-blue horseshoes), and paxillin (green ovals) is weakly associated with the actin filaments within the actin cytoskeleton. (**B**) When integrins engage with the ECM and when force is applied either intrinsically or extrinsically, talin is unfolded revealing cryptic vinculin-binding sites. Vinculin (brown ovals) is then recruited allowing for further engagement with actin filaments. Additionally, focal adhesion kinase (yellow circle) is recruited and activates downstream signals that culminate with increased non-muscle myosin contractility (light-blue ovals).

**Figure 4 biology-11-00052-f004:**
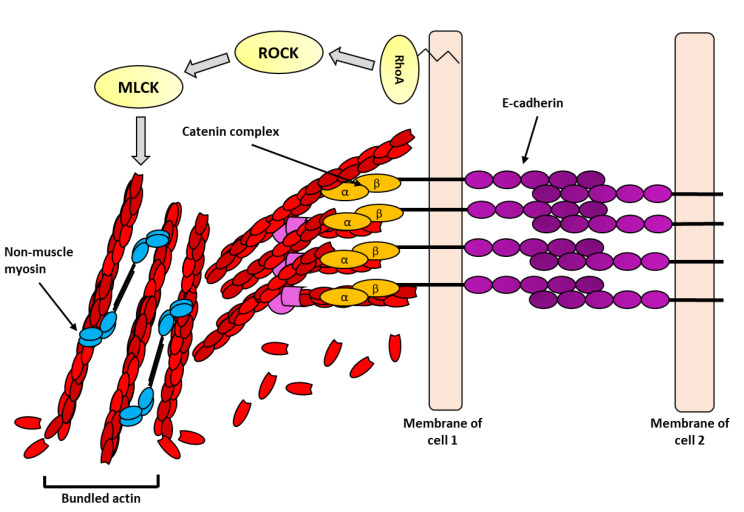
Hypothetical illustration of the actin cytoskeleton at adherens junctions under force: E-cadherin (purple circles) forms strong homophilic interaction between adjacent cells. Adaptor proteins, such as, β- and α-catenin form cytosolic adhesion complex (orange circles) and link E-cadherin to the actin cytoskeleton (red units). In response to force, the RhoA pathway (yellow) is activated, which in turn activates the actomyosin pathway culminating in force generation by non-muscle myosin (blue ovals) and further reinforcement of the actin cytoskeleton.

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
