# Peer review of "Dynamics of the Actin Cytoskeleton at Adhesion Complexes"

_biology, 2021, doi:10.3390/biology11010052_

Round 1

Reviewer 1 Report

This short overview of how actin is regulated by various proteins and by mechanical forces is clearly written and accessible. Although it could easily be expanded to include more actin regulators and (many) more references, I think it hits the highlights and is an accessible short review that will be of interest to students and others coming into the field.

I have a few suggestions and edits: 

44. Consider including myosin and formin in the actin bundling section in addition to their other functions. 

69. Clarify what the unique function of myosin in adhesions is. 

Section 1: Include a brief discussion of capping protein. It is in the figure but not, I believe, in the text. 

91. The concluding sentence should focus on the mechanism of force generation by actin polymerization - cellular function is too vague. 

104. Reduce wordiness to make the meaning of this sentence more clear. 

112. Consider rewording this statement. Because of cross-linking, and anchoring, actin filaments don't usually slide past each other (and 'slides past' does not evoke 'contract' to the reader). 

119 I recommend deleting 'Recent data shows'.

123 (and elsewhere) I recommend removing 'is suggested to be', 'was found that' and other filler phrases.

124. Specify to which mode of cell migration this is relevant.

180. Missing period at end of line.

186. You may wish to mention mechanical activation (tension activation) of alpha catenin binding to actin.

222. Reduce wordiness.

247. Adhesions should be adhesion

252. Sentence redundant with previous sentence.

266 Should be 'sites' .

Author Response

We thank the reviewer for his/her thoughtful comments which helped improved the overall quality of the review.  We have made all of the suggested modifications. Below is the point by point response: 

44. Consider including myosin and formin in the actin bundling section in addition to their other functions.   It has been added.

69. Clarify what the unique function of myosin in adhesions is. We clarified in the Introduction text that this information appears in a later section when myosins are discussed in detail.

Section 1: Include a brief discussion of capping protein. It is in the figure but not, I believe, in the text. Yes, you are correct.  We added capping protein to the text.

91. The concluding sentence should focus on the mechanism of force generation by actin polymerization - cellular function is too vague. Thanks for finding this overstatement.  We have corrected it. 

104. Reduce wordiness to make the meaning of this sentence more clear. Done

112. Consider rewording this statement. Because of cross-linking, and anchoring, actin filaments don't usually slide past each other (and 'slides past' does not evoke 'contract' to the reader). The statement has been re-worded.

119 I recommend deleting 'Recent data shows'. DOne

123 (and elsewhere) I recommend removing 'is suggested to be', 'was found that' and other filler phrases.  We searched the text for these terms and removed them. 

124. Specify to which mode of cell migration this is relevant.  It has been added.

180. Missing period at end of line. It has been added.

186. You may wish to mention mechanical activation (tension activation) of alpha catenin binding to actin.

222. Reduce wordiness. Done

247. Adhesions should be adhesion. The typo has been fixed.

252. Sentence redundant with previous sentence. It has been deleted.

266 Should be 'sites' . The typo has been fixed.

Reviewer 2 Report

This review is interesting, important and timely. Only minor corrections are needed at this point.

  • Fig. 1 is overlapped with the text.
  • "Filipodia" should be corrected to "Filopodia". If "filipodia" are distinct from filopodia, this needs to be stated clearly.
  • In the part regarding mechanotransduction at focal adhesions, work showing the effect of force on integrin receptors (Boettiger, Springer) needs to be discussed and referenced.
  • Please state the current scenario regarding force generation and mechanotransduction in 3D environments, and how adhesion dynamics are affected by friction vs integrin-dependent contacts (Paluch)

Author Response

We thank the reviewer for his/her comments as they have helped us improve the overall quality of the review. Below is a detailed response to the suggested modifications.  

This review is interesting, important and timely. Only minor corrections are needed at this point.  Thank you.

  • Fig. 1 is overlapped with the text. This formatting issue appears to have occurred after submission.  We will work with the journal to ensure proper formatting.   
  • "Filipodia" should be corrected to "Filopodia". If "filipodia" are distinct from filopodia, this needs to be stated clearly. Yes, indeed.  Thanks for catching this typo. 
  • In the part regarding mechanotransduction at focal adhesions, work showing the effect of force on integrin receptors (Boettiger, Springer) needs to be discussed and referenced. We added a discussion and the references.  
  • Please state the current scenario regarding force generation and mechanotransduction in 3D environments, and how adhesion dynamics are affected by friction vs integrin-dependent contacts (Paluch).  We added a discussion and the references.